# Recycling Chips of Stainless Steel Using a Full Factorial Design

**Claudiney Mendonça [1] , Patricia Capellato [1] , Emin Bayraktar [2,3,* ] , Fábio Gatamorta [2] , José Gomes [1] , Adhimar Oliveira [1] , Daniela Sachs [1] , Mirian Melo [1] and Gilbert Silva [1]**

1   Institute of Physics and Chemistry, Universidade Federal de Itajubá, Itajubá 37500-903, Brazil
2   Departamento de Engenharia de Manufatura e Materiais, Universidade Estadual de Campinas, FEM, Campinas 13083-860, Brazil
3   School of Mechanical and Manufacturing Engineering, Supmeca/Paris, 93407St Ouen, France
*   Correspondence: bayraktar@supmeca.fr; Tel.: +33-6-76-10-36-22

**Abstract:** The aim of this study was to provide an experimental investigation on the novel method for recycling chips of duplex stainless steel, with the addition of vanadium carbide, in order to produce metal/carbide composites from a high-energy mechanical milling process. Powders of duplex stainless steel with the addition of vanadium carbide were prepared by high-energy mechanical ball milling utilizing a planetary ball mill. For this proposal, experiments following a full factorial design with two replicates were planned, performed, and then analyzed. The four factors investigated in this study were rotation speed, milling time, powder to ball weight ratio and carbide percentage. For each factor, the experiments were conducted into two levels so that the internal behavior among them could be statistically estimated: 250 to 350 rpm for rotation speed, 10 to 50 h for milling time, 10:1 to 22:1 for powder to ball weight ratio, and 0 to 3% carbide percentage. In order to measure and characterize particle size, we utilized the analysis of particle size and a scanning electron microscopy. The results showed with the addition of carbide in the milling process cause an average of reduction in particle size when compared with the material without carbide added. All the four factors investigated in this study presented significant influence on the milling process of duplex stainless steel chips and the reduction of particle size. The statistical analysis showed that the addition of carbide in the process is the most influential factor, followed by the milling time, rotation speed and powder to ball weight ratio. Significant interaction effects among these factors were also identified.

**Keywords:** duplex stainless steel; chip; high-energy milling; particle size; factorial design

## 1. Introduction

Duplex stainless steels (DSS) are two-phase austenite ($\gamma$-CFC) and ferrite alloys which are capable of combining the good properties of ferrite and austenitic stainless steels [1–4]. Duplex stainless steels are mainly used in the following industries: pulp and paper, disinfection of plants, gas flue cleaning, heat exchangers, and the nuclear and chemical industries [5–7]. Currently, stainless steel production is one of the most important and fastest growing metallurgical industries in the world. The world production of stainless steel reached approximately 45.8 million tons in 2016, an increase from the 41.5 million tons produced in 2015 [8].

Due to the applicability of this material, its reuse is necessary, but its reuse through the casting process is expensive, due to the subsequent thermomechanical processes carried out. The use of the high energy milling process is an alternative for the reuse of this material. Reutilization of recycled materials has as a main objective the reduction of environmental impact and rationalization of energy chains [9]. Stainless steel components produced by powder metallurgy comprise an important and

growing industry segment [10–13]. This process generates few residues, in addition to the possibility of reusing the raw material in the process, engendering benefits for the environment [9]. Duplex stainless steel obtained by powder metallurgy technology could be used in many industry segments due to their mechanical properties and good corrosion resistance [14].

Studies have been carried out to produce composites of a sintered metal matrix with the addition of ceramic particles, called oxides or carbides, such as vanadium carbide (VC), niobium carbide (NbC), titanium carbide (TiC) and tungsten carbide (WC). These particles have attracted interest due to their exceptional mechanical, physical and chemical properties, aiming to increase the high energy milling efficiency and the mechanical strength of the material [15,16]. Vanadium carbide is an important material for industrial applications due to excellent resistance to high temperatures, and its high chemical and thermal stability, even at high temperature [17]. In some studies, it was verified that the addition of carbide in the process of the high energy milling of chips from the machining process, resulted in greater efficiency of the milling process, with a greater decrease in the size of the particles, obtaining sizes of nanometric particles [18–20].

In the milling process, the particle size is an important factor in stainless steel processed by powder metallurgy, which affects its compressibility, increasing the densification and properties of sintered products. So, given these critical aspects for the final powder quality, experimental approaches that allow a wide investigation of the milling process and stablish mathematical relationships among its variables are important and necessary. In this context, the full factorial designs are experimental planning considered in the literature as suitable methods for this aim [21]. By setting only two levels for each control variable of the process, this approach is able to rank the more statistically significant variables on the analyzed results, to estimate their main and interaction effects, and to represent this process behavior by a mathematical model. Furthermore, the planned levels ensure balanced combinations in relation to the experimental variations, making possible a good understanding about the process behavior as from consistent information obtained with a minimum number of experiments. However, even being able to analyze non-linear relationships for the process variables, the factorial models are truncated in first order polynomials.

Shashanka & Chaira developed a duplex stainless steel in a Fritsch planetary mill with 40 hours duration and a rotation speed of 300 rpm [22]. Rahmanifard et al. studied the effect between a powder to ball weight ratio of (10:1 and 15:1) and a rotation speed between (300 and 420 rpm) during the ferrite stainless steel milling [23]. The authors observed that crystallite sizes, and the ferritic stainless steel powder particle sizes, reduced when rotation speed and powder to ball weight ratio were increased. Pandey et al. prepared ferritic stainless steel with different powder to ball weight ratios (10:1, 15:1 and 20:1) and rotation speeds (250, 300 and 350 rpm) [24]. They reported that crystallite size and particle sizes of ferritic stainless steels decreased as powder to ball weight ratio and rotation speed increased.

Thus, it is possible to verify the great importance of establishing parameters such as rotation speed, milling time, powder to ball weight ratio and carbide percentage in the milling of stainless steel duplex. Although not much literature is available on the effect of milling parameters, such as the effect of ball-powder ratios and the effect of rotation speeds during synthesis of ferritic stainless steel by planetary milling and stainless steel duplex [23]. In other papers, we verified the influence of time and carbide addition to chip milling of an aluminum bronze alloy [21] and a stainless steel [4].

Based on these considerations, the aim of this research is to develop an experimental study on the influence of high-energy milling parameters in duplex stainless steel chips processing, which represents a sustainable alternative for the efficient reuse of these chips. For this, the presented statistical analysis and modeling were performed as from a full factorial design, planned for four investigated factors. The choice by the factorial planning while experimental strategy is justified by the reasons previously mentioned [23,24]. The production of duplex stainless steel powders with the addition of carbides by high-energy mechanical milling is a novel method for recycling chips.

## 2. Materials and Methods

In this study, the UNS S31803 duplex stainless steel has the following chemical composition: 22.3% Cr; 5.44% Ni; 2.44% Mo; 0.02% C, 0.160 N and Fe bal. The raw material for the research was obtained by the machining step at low speed, and without the use of lubricants, to avoid contamination by oil-soluble. Via the procedure described, we obtained the UNS S31803 stainless steel in the form of scraps which were subsequently used in the milling process. The average proportion of the alloy chip sizes was described using a stereoscope. At the starting point of the milling process, we used of duplex stainless steel chips, UNS S31803, with and without VC present. The initial chip sizes were between 5 and 15 mm.

For the chip milling process, we used high-energy milling in a planetary ball mill (Yangzhou Nuova Machinery Co., LTD, Yangzhou, China) with inert atmosphere of argon. Analysis of variance (ANOVA) using the Minitab program (Minitab 15, Minitab, LLC Minitab® 15 Statistical Software, State College, PA, USA) evaluated the factors significance and their interaction. Thus, according to the DOE (Design of experiments) approach, where the number of experiments was determined by a $2^n$ full factorial design, there were varied "$n = 4$" parameters, which generated 16 experiments configured into two levels. For a greater reliability of the results, two replicates were made, resulting in 32 experiments.

Through the use of an experimental design, we consistently determined the optimal condition for "smallest particle size". The milling parameters to be varied were: rotation speed, milling time, powder to ball weight ratio and vanadium carbide percentage. Values were chosen based on the literature, and mainly, by preliminary empirical tests performed in high-energy milling. These values levels are in Table 1. As previously mentioned, the experimental combinations into two levels are inherent characteristics for the factorial planning. If these levels are continuous variables, the internal behavior among them can be estimated and analyzed in a satisfactory way, by means of a non-linear first order polynomial and with a minimum number of experiments.

**Table 1.** Milling parameters and their levels

| Factor Investigated | Unit | Notation | Levels | |
| --- | --- | --- | --- | --- |
| | | | −1 | +1 |
| Rotation speed | rpm | *Rs* | 250 | 350 |
| Milling time | h | *Mt* | 10 | 50 |
| Powder to ball weight ratio | - | *Pr* | 10:1 | 20:1 |
| Carbide percentage | % | *Cp* | 0 | 3,0 |

We used a particle size analyzer (brand: Malvern Master Sizer (Malvern Master Sizer: Microtrac model S3500, Microtrac Global Location, Montgomeryville, PA, USA); model: 2000) to determine the distribution of particle sizes. For morphology characterization, size identification, and particle distribution, we used a scanning electron microscope (SEM), brand: Carl Zeiss and EVO (U.S. Carl Zeiss Company (Carl Zeiss & EVO), New York, NY, USA), model: MA15, secondary electron mode.

## 3. Results and Discussion

Initial characterizations of duplex stainless steel chips UNS S31803 are illustrated in Figure 1. It is possible to observe that machined stainless steel chips have an average size of 8 mm (Figure 1), and regions of surface plastic deformation, caused by the machining tool, are observed.

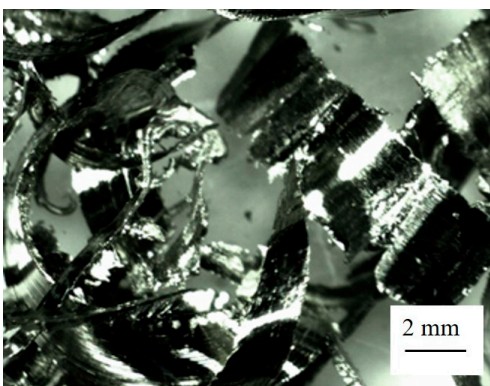

**Figure 1.** UNS S31803 stainless steel duplex scrap.

The effect of a factor is defined as the change in response produced by a change in the level of the factor. This is frequently called a main effect as it refers to the primary factors of interest in the experiment [23].

The mathematical model for factorial planning $2^4$ is given by Equation (1), where $R$ is the average size of the particle and $Rs$, $Mt$, $Pr$, $Cp$ mean rotation speed, milling time, powder to ball weight ratio and carbide percentage, respectively. For the following mathematical model, all the coefficients were estimated in its coded format, as from the experimental results and using the Ordinary Least Square method (OLS), this latter present in the statistical software Minitab.

$$\begin{aligned}
R = {} & 248,695 - 58,845 \cdot Rs - 69,849 \cdot Mt - 45,737 \cdot Pr - 71,461 \cdot Cp - 6014 \cdot Rs \cdot Mt \\
& + 19,698 \cdot Rs \cdot Pr + 4999 \cdot Rs \cdot Cp - 2068 \cdot Mt \cdot Pr + 18,595 \cdot Mt \cdot Cp - 4006 \cdot Pr \cdot Cp \\
& + 8717 \cdot Rs \cdot Mt \cdot Pr + 5730 \cdot Rs \cdot Mt \cdot Cp - 1271 \cdot Rs \cdot Pr \cdot Cp + 4688 \cdot Mt \cdot Pr \cdot Cp \\
& - 5577 \cdot Rs \cdot Mt \cdot Pr \cdot Cp
\end{aligned} \tag{1}$$

In addition, the model presented an adjusted square correlation coefficient $R^2$ (adj) of 93.27%, fitting the statistical model quite well. Table 2 shows the average particle size values for each condition stipulated in the experimental design for the variables rotation speed, milling time, powder to ball weight ratio and carbide percentage.

**Table 2.** Average particle sizes of duplex stainless steel for the investigated milling parameters (experimental matrix in coded variables).

| Experiment | Rotation Speed | Milling Time | Powder to Ball Weight Ratio | Carbide Percentage | Replicate 1 | Replicate 2 |
|---|---|---|---|---|---|---|
| | (rpm) | (h) | - | (%) | (μm) | (μm) |
| 1 | −1 | −1 | −1 | −1 | 508.00 | 496.70 |
| 2 | +1 | −1 | −1 | −1 | 380.30 | 389.30 |
| 3 | −1 | +1 | −1 | −1 | 366.30 | 415.80 |
| 4 | +1 | +1 | −1 | −1 | 187.60 | 151.10 |
| 5 | −1 | −1 | +1 | −1 | 394.30 | 443.80 |
| 6 | +1 | −1 | +1 | −1 | 354.10 | 302.30 |
| 7 | −1 | +1 | +1 | −1 | 267.60 | 179.50 |
| 8 | +1 | +1 | +1 | −1 | 143.90 | 141.90 |
| 9 | −1 | −1 | −1 | +1 | 300.90 | 398.50 |
| 10 | +1 | −1 | −1 | +1 | 246.50 | 177.50 |
| 11 | −1 | +1 | −1 | +1 | 229.40 | 268.20 |
| 12 | +1 | +1 | −1 | +1 | 109.70 | 85.11 |
| 13 | −1 | −1 | +1 | +1 | 222.20 | 206.60 |
| 14 | +1 | −1 | +1 | +1 | 163.90 | 111.80 |
| 15 | −1 | +1 | +1 | +1 | 90.54 | 132.30 |
| 16 | +1 | +1 | +1 | +1 | 48.91 | 43.68 |

Table 3 presents the analysis of variance for the full $2^4$ factorial design with two replicates. The data indicate that the main effects of rotation speed, milling time, powder to ball weight ratio and carbide percentage are significant for the decrease of particle size. The interaction between rotation and ratio, as well as the variables time and carbide, are significant, because the value of $p$-Value is lower than the significance level adopted at 5% probability level ($p < 0.05$).

**Table 3.** Analysis of variance for medium size (µm).

| Source | DF | Adj SS | Adj MS | F | p |
|---|---|---|---|---|---|
| Main Effects | 4 | 497,285 | 124,321 | 104.65 | 0.000 |
| 2-Way Interactions | 6 | 26,088 | 4348 | 3.66 | 0.018 |
| 3-Way Interactions | 4 | 4237 | 1059 | 0.89 | 0.491 |
| 4-Way interactions | 1 | 995 | 995 | 0.84 | 0.374 |
| Residual error | 16 | 19,008 | 1188 | - | - |
| Total | 31 | - | - | - | - |

$S = 34.47$, $R_{sq} = 96.53$, $R_{sq(adj)} = 93.27$.

The other second-order, third-order and fourth-order interactions are not significant, as they present values above the significance level of (5%). In addition, the model guarantees a good correlation ($R^2 = 96.53\%$), if by adjusting the statistical model ($R^2(adj) = 93.27\%$), these values represent the percentage of data observed in the response that the mathematical model can explain.

Figure 2 shows the residual analysis for the particle size response, which is characterized as an important procedure to ensure that the mathematical models developed consistently represent the responses of interest. A normal probability graph is only a graph of the cumulative distribution of residual on normal probability plot, which is the graph with the ordinates staggered so that cumulative normal distribution is potted as a straight line [25]. In addition, the normal probability plots of the residuals must be distributed normally and independent of each other. The points in the normal probability plots of the residuals reveal a straight line confirming the reasonableness of the model. Normality test plot and histogram indicated normal distribution of residuals. On the other hand, the residual graphs standardized by intensity of the responses and by order of the experiments all indicate a random dispersion of the residuals (as can be seen in Figure 2).

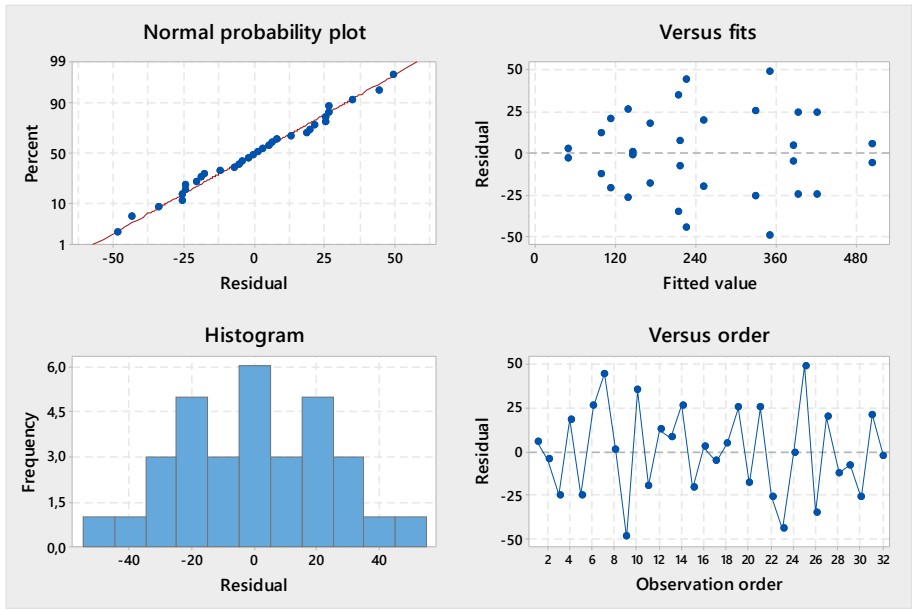

**Figure 2.** Residual analysis of chips for particle sizes.

Statistically significant effects can be observed with the help of the Pareto diagram (Figure 3). The factors or interactions outside of the dotted line in 2.12 are significant in decreasing order: the carbide percentage, milling time, rotation speed, powder to ball weight ratio, *Mt* and *Cp* interaction and finally the interaction between *Pr* and *Rs*. The carbide percentage was more influential in the milling process. This material in contact with the stainless steel chips at the time of milling aids the milling process, increasing the efficiency of the process.

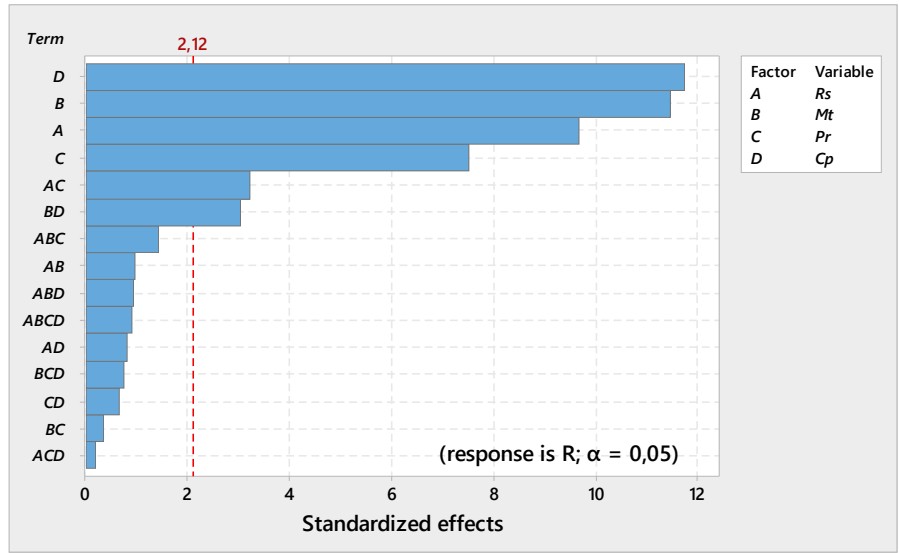

**Figure 3.** Pareto graph analysis for the factorial experiment with 4 levels.

Figure 4 shows the graph of the main effects of the investigated factors in relation to particle size. In the graph of the main effects, it is inferred that a factor is directly related to the length and slope of the line in the graph of Figure 4 [26]. The larger the slope, the higher the influence on the decrease of the average particle size when changing from a low level to a high level. So, once these main effects are derived from a statistical adjustment of 93.27%, with a p-value less than 5% of significance (which represent a confidence level of 95%), these results are reliable for this milling process.

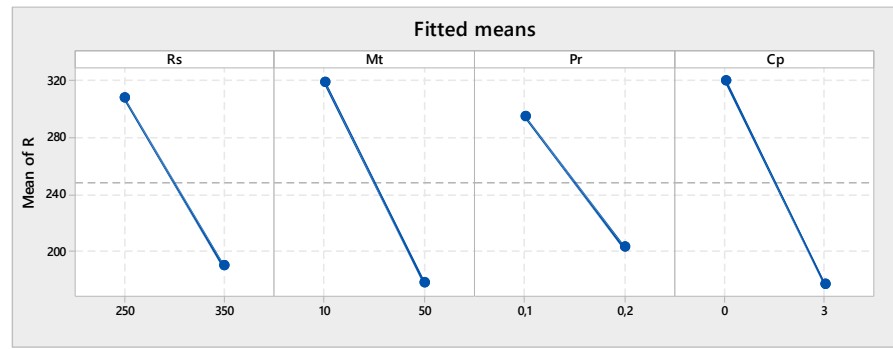

**Figure 4.** Main effect analysis of milling parameters: *Rs*, *Mt*, *Pr*, *Cp* for the experimental design.

The greatest influence on the reduction of particle size is the variation of the amount of vanadium carbide added to the milling process, which presents a greater slope and length of the curve. That is, when a minimum level change occurs (addition of 0% of carbide) to the maximum level (addition of 3% vanadium carbide), this change becomes more significant for particle size reduction, as explained above. Other parameters are also influential: in descending order are milling time, rotation speed and powder to ball weight ratio.

The effects analyzed are negative, a reduction in particle size is observed when the factor goes from the lowest level to a higher level. The percentage of carbide and milling time are the most important parameters for decreasing particle size.

According to Suryanarayana the milling time is the most important parameter in the milling process [27]. As a general rule, it may be appreciated that the times taken to achieve steady-state conditions are short for high-energy mills and longer for low-energy mills. The increase of the milling time is characterized by the increase of the plastic deformation, generated by the impacts of the milled bodies, by which it leads to fracturing (due to the brittleness induced by the hulling) [4,20,27–32].

Rotation is an important parameter when it comes to milling efficiency (i.e., the frequency and kinetic energy of shocks generated during milling) [27]. However, for low rotations the frequency and kinetic energy of shocks generated during milling are not sufficient for a reduction of the particle size.

An interaction (Figure 5) is effective when the change in response from the lowest to the highest level of a factor is dependent on the level of a second factor, (i.e. when the lines are not parallel) [29]. Thus, the combination of the parameters or the interaction between the time variable and the carbide percentage is also significant, as well as the interaction between the rotation and the ratio variables, since the lines of the interaction graph are not parallel to each other. The higher the rate of rotation and the mass/sphere ratio, the lower the particle size value obtained, due to the higher energy involved in the process, which causes a greater particle breakage and a subsequent decrease in particle size.

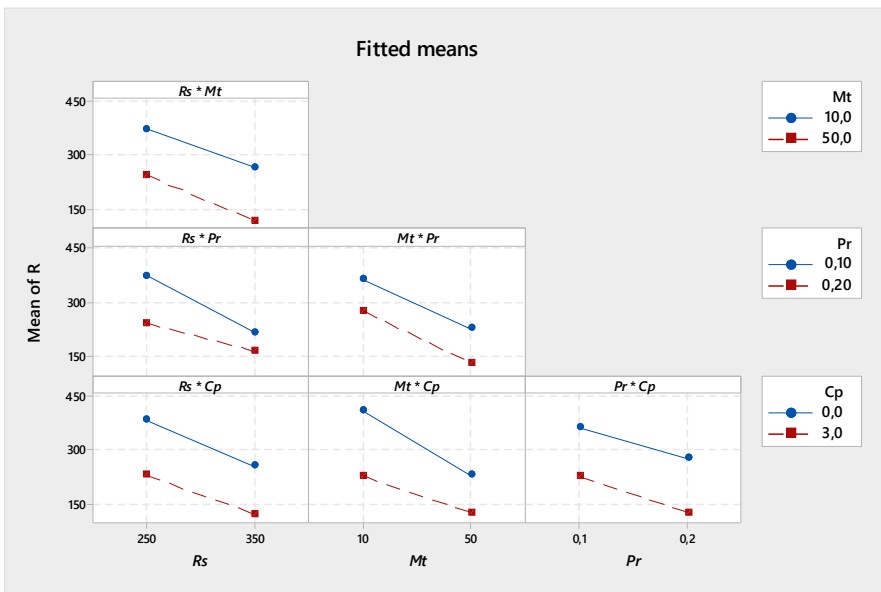

**Figure 5.** Interactions effects analysis among milling parameters: *Rs*, *Mt*, *Pr*, and *Cp*.

With the increase in rotation and the mass/sphere ratio, the higher the energy applied to the grinding of the powder, which favors milling and particle size reduction. The higher mill speed increases the impact energy of balls and thus increases the rate of collision between ball-powder-jar. Shashanka and Chiara, when milling and producing the duplex stainless steel, verified that mean particle size decreased from 77 to 15 μm during a milling from 0 to 40 h [25]. Dias, et al., with increased chip milling time of an aluminum bronze alloy, found that carbide addition and milling time are the parameters that most influence the decrease in particle size [25].

The cube plot (Figure 6) shows the interactions between the factors and the responses obtained for the particle size for each experiment performed. It can be observed that for all the analyzed parameters the average particle size decreases with the addition of vanadium carbide. For the purpose of comparison, it can be seen that for the process with the addition of 3% of vanadium carbide, a mass/ball ratio of 1/20, and a rotation of 350 rpm, the obtained particle size was smaller than the grinding performed for 50 h, with the same conditions of the previous milling, but without the addition

of carbide. In that case, the value decreased from 142.9 μm to 43 μm. The efficiency of the milling process increased significantly with the addition of vanadium carbide in the process.

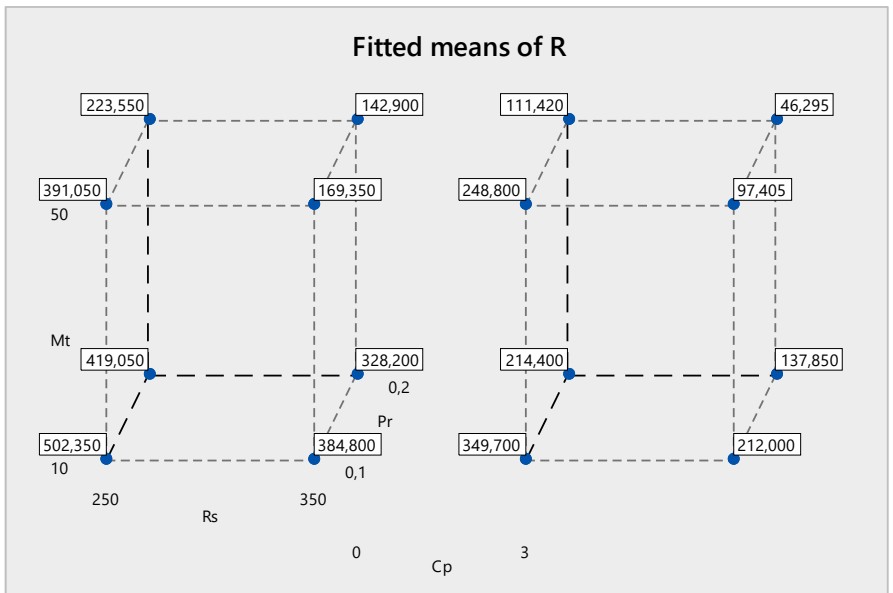

**Figure 6.** Cube plot for the average particle size in the full factorial design.

The mass ratio of the material to the ball's mass is an important parameter for achieving the desired structure or grain size for the ground material. Thus, if this ratio is higher, the shorter the time required for milling, as the number of collisions increases at each instant and, consequently, increases the internal energy of the milling system [29–33]. In Figure 6 it is observed that the particle size decreases with increasing milling time and with the ball mass ratio for grinding without addition of carbide. However, these parameters become less important with the presence of the carbide in the milling, as it is possible to check in the figure.

As previously described, a correlation was found between the mass/ball ratio and rotation variables and also a relationship between carbide percentage and milling time. Thus, for a better analysis of the results, the response surface analysis were performed. Two surface plots of the regression model are shown in Figure 7, in which two parameters were kept constant at their upper level and the other two vary within the experimental intervals. The surface plots of the response functions are useful in understanding both the main and interaction effects of the factors [32,33].

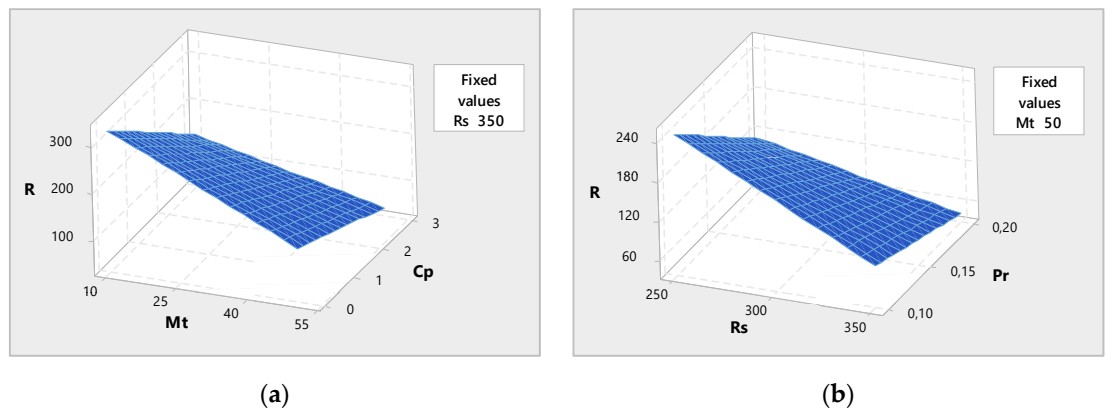

(**a**) (**b**)

**Figure 7.** Response surface graphs of particle size: (**a**) milling time versus carbide percentage; (**b**) rotation speed versus powder to ball weight ratio.

The particle size variation can be analyzed by the plots of surface response (Figure 7a) in which two variables are kept constant (rotation and relation) while the amount of carbide (0 to 3 wt.%) and time (10 h and 50 h) are variables. With an increase in milling time, the particle size decreases. This behavior is observed for both grinding with and without carbide addition. The work hardening causes powders to become brittle, and the fracturing process becomes significant for longer milling times [30].

With the increase of carbide in the grinding process the particle size also decreases, being that for milling with the addition of 3% VC, the particle size obtained presents the same order of magnitude for milling for 50 h, without the addition of carbides. Canackçi & Varol, found a decrease in particle size for the chip milling of an aluminum alloy [32]. What's more, Dias et al., obtained the smallest particle sizes for high milling times [29]. These authors verified that the time of milling and addition of carbides are the most influential parameters in the milling process.

Figure 7b depicts the analysis of particle size behavior for the variation of the ratio (1/10–1/20) and rotation (250–350 rpm) parameters, of which the values are 3% carbide percentage and 50 h milling time. The effects analyzed are negative, because with the increase of the parameters of rotation and mass/ball ratio the particle size decreases. For rotations near 250 rpm and mass/ball ratio around 1/10, the particle size obtained is larger than 200 µm. The higher mill speed and mass/ball ratio, the higher the energy of balls, thus increases the rate of collision between ball-powder-jar. Consequently, with the increase of the impacts, there is a greater reduction of the particles. The greater the speed, the balls can fall down from the maximum height to produce the utmost collision energy [28,33].

Figure 8a,b depict the SEM micrographs of 50 h milled duplex stainless steel powders with and without vanadium carbide. It can be seen in Figure 8a that, the UNS S31803 stainless steel in the form of chips was transformed into particles with irregular morphology and sizes ranging from 85 to 250 µm with an average particle size of 151.8 µm after milling at 350 rpm, 50 h milling, and a mass/ball ration of 1/10, without the addition of vanadium carbide. For a milling of the material with carbide, it was found that the material acquired an irregular morphology with an average size of 45 to 140 µm and an average particle size of 77.9 µm, as can be seen in Figure 8b.

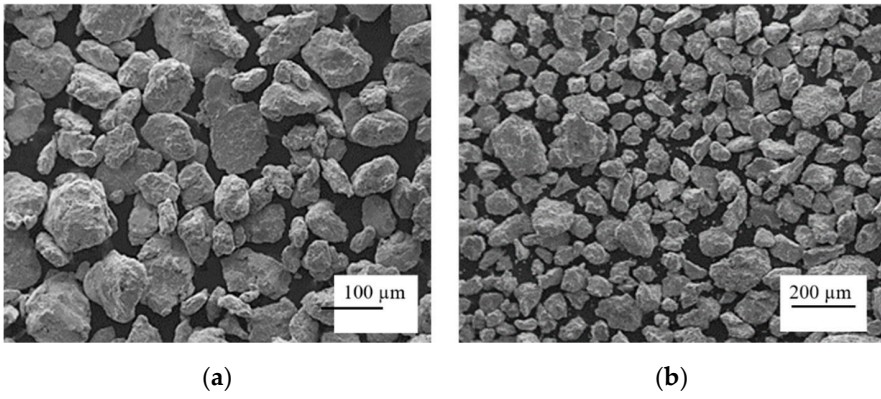

(**a**)          (**b**)

**Figure 8.** Photomicrograph of stainless steel powders: (**a**) milling at 350 rpm, mass/ball ratio of 1/10, milling time of 50 h; (**b**) milling at 350 rpm, mass/ball ratio of 1/10, milling time of 50 h with addition of 3% of vanadium carbide.

Addition of carbide in the milling process increased the efficiency of the milling process, with a decrease of 48.7% of the particle size. The SEM analysis indicates the same behavior observed in the analysis of the pareto graphs and interactions. Reduction mechanism may be due to the incorporated carbides, which act as concentration agents of internal stresses, which are produced by the collisions in the milling process. Dias, et al., 2018, found that when milling with vanadium carbide for 50 h the particle sizes obtained were in the order of 50 µm. Mendonça, et al. found that the addition of carbide in the milling of chips of the same stainless steel results in a reduction in particle size of the order of 20% when compared without carbide [28].

The reduction mechanism was due to the incorporated carbides, which act as concentration agents of internal stresses, which are produced by the collisions in the milling process. As the process of collisions is continuous, the particles of the material are increasingly hardened and the level of tension rises to the point that, where the carbides are located nucleation of cracks occurs, fracturing the material, thus reducing the particle size, as shown in Figure 9.

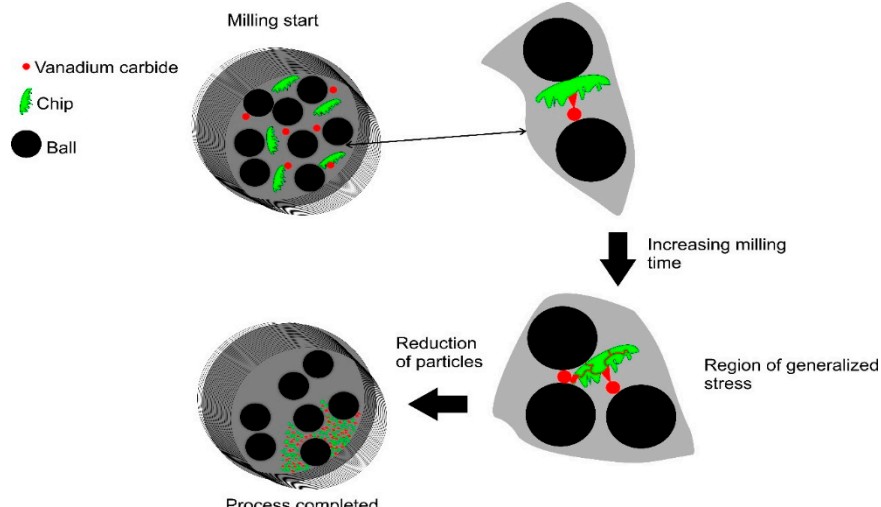

**Figure 9.** Mechanism of milling with addition carbide of the stainless steel duplex chips.

A smaller particle size was obtained for a more aggressive milling condition, which was 350 rpm, with a mass/ball ratio of 1/20, for 50 h, with addition of 3% VC (Figure 10). It is observed that the material acquired an acicular morphology with an average size of 25 to 135 μm. In this process, a greater amount of energy is involved then when allied with the carbide addition, which favors the reduction mechanism of the particle size. The smallest particle size obtained after 50 h of milling with 3% VC is 174 times smaller than the chip.

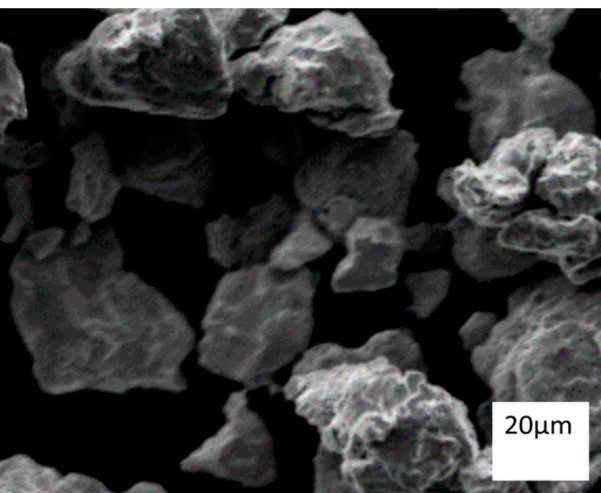

**Figure 10.** Photomicrography of stainless steel powder of milling parameters of 350 rpm, mass/ball ratio of 1/20, milling time of 50 h.

## 4. Conclusions

The high-energy milling process is an alternative route to the reuse of stainless steel chips with and without addition of VC.

The results showed with the addition of carbide in the milling process cause an average of reduction in particle size when compared with the material without carbide added. Through the analysis performed in this study, it was possible to verify that with the addition of carbides in the milling process, there was an average reduction in particle size when compared to the material without carbide added; the difference was around 66%.

Also, all the four parameters applied in this study influenced the process of milling of duplex stainless steel chips and the reduction of particle size. However, the statistical analysis showed that the addition of carbide in the process is the most influential factor, followed by the milling time, rotation speed, and powder to ball weight ratio. The production of duplex stainless steel powders with the addition of carbides by high-energy mechanical milling is a novel method for recycling chips.

**Author Contributions:** C.M., J.G., investigation; F.G., G.S., resources; C.M., data curation; C.M., writing—original draft preparation; P.C., J.G., E.B., writing—review and editing; A.O., D.S., visualization; J.G., supervision; J.G., project administration; M.M., funding acquisition.

**Funding:** This research received no external funding.

**Acknowledgments:** The authors thank the Brazilian agencies, CAPES, CNPq and FAPEMIG, for their financial support of the graduated students.

**Conflicts of Interest:** The authors declare no conflict of interest.

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
