# Peer review of "Recycling Chips of Stainless Steel Using a Full Factorial Design"

_metals, doi:10.3390/met9080842_

Round 1
Reviewer 1 Report
I still keep my opinion, that it is a contradiction between the equation (1) and experimental results.
D has a positive sign, so VC percentage increases the particle size based on the equation (1).
Author Response
Reviewer-I
Dear reviewer, thank you for your kind critics and suggestions that we have revised again according to points that you have indicated;
1- English language and style and minor spell check have been made
2- Introduction has been improved and added certain references
3- And also the conclusion has been improved regarding the new revision.
4- Question about a contradiction between the equation (1) and experimental results. D has a positive sign, so VC percentage increases the particle size based on the equation (1).
Thank you again for this remarkable point that you have made.
We have again corrected this equation and the experimental results do show the correct agreement with this equation (1).

Reviewer 2 Report
Dear Authors,
Congratulations on your work, which has been deeply improved.
However, some improvements need yet to be performed. Please pay attention to the following comments and suggestions:
Please avoid the use of a large number of references for a single idea ([1-4], [5-7].
Although the raw material used in this work are chips generated in Machining processes, no reference is made about the Machining process os Duplex Stainless Steel alloys, mainly from MDPI journals. Please improve the Introduction, adding some important paper in this field, like:
Coatings 2016, 6, 51; doi:10.3390/coatings6040051
Coatings 2019, 9, 392; doi:10.3390/coatings9060392
https://doi.org/10.1007/s00170-019-03351-8
In line 75, the reference "Shashanka and Chaira seems to be wrongly used. This reference is the [18], but the sentence contains the reference [20]. This reveals lack of care. Please pay attention to these details, which are very important to the reader. Moreover, include the reference immediately after the name of the authors referred.
The Materials and Methods section needs to be deeply improved. The Authors are mainly focused in the statistical analysis, but the details about the process have too much importance to be ignored. Thus please explain why did you use low speed machining to obtain the chip. Moreover, please define the machining process used to produce the chips.
In order to allow the reader realize how the chips are, more details about the machining process need to be given, namely the machining parameters used in the process (chip cross section area, feed rate, depth of cut, cutting speed, and tool type used). Is the tool provided with chip breaker?
Please include more details about the mechanical properties of the DSS alloy, namely the UTS and the hardness.
In line 103, when you use the term "scraps", I think you would like to use "chips", right?
Please avoid the use of the first person in scientific documents ("we")..
The English/Sentence construction needs to be improved in line 104: "At the starting point of the milling process, we used of duplex stainless steel chips...with and without VC present".
Please explain what means "high energy milling".
Please explain what means "powder to ball weight ratio".
After this, the work is too much focused in the statistical analysis, loosing some focus in the practical results really obtained.
Please explain the sentence in line 310 (Conclusions): "Through the analysis performed in this study, ...was around 66%".
Please let the readers now the final destination of the duplex stainless steel powders generated by this process.
Good luck.
Kind regards,
Author Response
Reviewer-II
Dear reviewer, thank you for your kind critics and suggestions. We have really appreciated that you have made critics and suggestions and also a correction.
We have revised and corrected all of the points that you have indicated for correction.
First of all;
1- English language has been checked and improved and certain typing errors have also been corrected
2- Introduction, main text and also the conclusion have been improved regarding the new revision
All other corrections and reconstruction of the tables and other points have been done and shown in red colours in the main text. Thank you again for this remarkable points that you have made.

Round 2
Reviewer 2 Report
Dear Authors,
Thank you for addressing my comments and suggestions.
Please pay attention to the new added references, because they are not numbered and all the remaining references should be updated in terms of numbering.
You must pay attention to the details.
Kind regards,
Author Response
Reviewer-II second round
Dear reviewer, thank you again for your kind attention and suggestions. We have really appreciated it. I am so sorry for the arrangement of the new references added in the introduction and the reference list.
We have revised again and corrected all of the points carefully with new spell check program.
Finally;
1- English language has been checked by my colleague having native language in English and improved and certain typing errors have also been corrected
2- Introduction, main text have been corrected regarding the new revision the arrangement of the new references added in the introduction and all of the update of the references in the text and also in the reference list.
All other corrections and reconstruction of the tables and other points have been done and shown in red and yellow colours in the main text. Thank you again for this remarkable points that you have made.
Kind Regards
Emin BAYRAKTAR
Corresponding author

Round 3
Reviewer 2 Report
Dear Authors,
Thank you só much for having properly addressed my comments and suggestions.
Good luck/Boa sorte,
Kind regards,
FGS
This manuscript is a resubmission of an earlier submission. The following is a list of the peer review reports and author responses from that submission.
Round 1
Reviewer 1 Report
Dear Authors,
Congratulations on your work, which is very well conducted. There are a few things which need to be more detailled in order to better explain the work carried out. It is in that sense that I'm providing below some suggestions in order to improve your work:
As Portuguese native speakers, there are some gramatical mistakes along the manuscript. Try to use some grammatical help software, like GRAMMARLY or HEMINGWAY EDITOR, which can help you to eliminate these mistakes. Indeed, the texto needs to be proof-read. I would like to help you, but there are several mistakes.
The number of references is good, but more than 50% of them have more than 10 years old. Please try to refresh the references, using recente papers (like https://doi.org/10.1007/s00170-019-03351-8), as well as papers from the publishing house where you are trying to publish (MDPI) (Coatings 2016, 6, 51; doi:10.3390/coatings6040051).
Please describe the reasons behind your selection of just two levels in the parameters studied. In that way, just with two values, it is difficult to confirm if the trend for each variable is solid. I know that this leads to a lower number of experiments, but the work suffers a lack of consistency in the results.
The use of parameters too close each other creates problems in terms of statistical analysis. Thus, please explain the main reasons that have driven you to select those parameters (250 and 350 rpm). These parameters should be selected based on previous works. Was this selection based on works carried out by other authors? Which are?
Please point out the type of matrix used in the DOE process (Lxx?).
In line 118, please explain how the expression (1) has been obtained. I think it was obtained from the Minitab software, but the readers need to know how to replicate the experiences and need to know the equation origin.
Figure 4 shows the trends regarding the variables. However, are this trend consistente? Sometimes, the trend between to values is not followed when the variable assumes other diferente value.
Between lines 257 and 269, the "reduction" idea is repeated. Please try to avoid this.
The discussion on how the carbide influences the size of the particles is very well done.
Once again, congratulations on your work.
Kind regards
Reviewer 2 Report
This article focuses on statistical analysis and not mechanical milling. As one can read in a review article: ’Processes inside planetary ball mills are complex and strongly depend on the processed material and synthesis and, thus, the optimum milling conditions have to be assessed for each individual system.’ (Christine Friederike Burmeister and Arno Kwade DOI: 10.1039/c3cs35455e)
So, this article is very strange in my opinion. Relying on just a couple statistics in mechanical milling (which as mentioned above, is a strongly complex process) is quite unfounded.
Both fracturing and cold-welding exist during mechanical milling. Some periods cold- welding dominates and other times fracturing. However, after milling for a certain time, steady-state equilibrium is attained.
In line 22-23: The results showed with the addition of carbide in the milling process occur an average of reduction in particle size when compared with the material without carbide added.
In line 116-117:‘The mathematical model for factorial planning 24 is given by Equation (1), where R is the average size of the particle and A, B, C, D mean rotation, rotation time, mass/sphere ratio and VC percentage, respectively.’ In the equation (1) D has a positive sign, so VC percentage increases the particle size. This is an enormous contradiction.
In line 89: ‘The initial chips sizes were between 5 and 15 mm.’
In line 109: machined stainless-steel chips have an average size of 8000 μm.
Another contradiction is found in connection with the chips size. There is a significant difference between 5 and 8000.
The Authors did not use enough technical terminology. I therefore think that the referred literatures should be studied better -rotation speed and not rotation, powder to ball weight ratio and not mass/sphere ratio, etc…
Some references are doubled: Refer 9 is the same as refer 10, and again refer 20 is the same as refer 23.